# Microbial biofilms for electricity generation from water evaporation and power to wearables

Xiaomeng Liu [1], Toshiyuki Ueki[2], Hongyan Gao[1], Trevor L. Woodard[2], Kelly P. Nevin[2], Tianda Fu [1], Shuai Fu[1], Lu Sun [1], Derek R. Lovley [2,3] ✉ & Jun Yao [1,3,4] ✉

Employing renewable materials for fabricating clean energy harvesting devices can further improve sustainability. Microorganisms can be mass produced with renewable feedstocks. Here, we demonstrate that it is possible to engineer microbial biofilms as a cohesive, flexible material for long-term continuous electricity production from evaporating water. Single biofilm sheet (~40 μm thick) serving as the functional component in an electronic device continuously produces power density (~1 μW/cm$^2$) higher than that achieved with thicker engineered materials. The energy output is comparable to that achieved with similar sized biofilms catalyzing current production in microbial fuel cells, without the need for an organic feedstock or maintaining cell viability. The biofilm can be sandwiched between a pair of mesh electrodes for scalable device integration and current production. The devices maintain the energy production in ionic solutions and can be used as skin-patch devices to harvest electricity from sweat and moisture on skin to continuously power wearable devices. Biofilms made from different microbial species show generic current production from water evaporation. These results suggest that we can harness the ubiquity of biofilms in nature as additional sources of biomaterial for evaporation-based electricity generation in diverse aqueous environments.

Sustainable strategies for energy production are required to reduce reliance on fossil fuels and to power electronics without generating toxic waste[1–6]. As ~50% of the solar energy adsorbed on earth drives evaporation[7], generating electricity from water evaporation through engineered materials is a promising approach[8,9], but power outputs have been low and the materials employed were not sustainably produced.

Specifically, evaporation-driven water flow at water-solid interfaces can drive charge transport for electricity generation[8,9]. In order to be effective, this streaming mechanism requires a material with a large surface area with associated mobile surface charges. Practicality will require low-cost materials and a minimum of processing. However, to date evaporation-based electricity generation has primarily focused on devices fabricated from thin films of assembled nanomaterials to improve the surface area, and further functionalized to introduce hygroscopic surface groups[8,9]. Furthermore, power densities and stability have been low. For example, an initial device based on functionalized carbon black achieved an energy density in the range of tens of nW/cm$^2$ for days[10]. A silicon-nanowire material increased power density close to μW/cm$^2$ level under the ambient environment[11], but the

[1]Department of Electrical Computer and Engineering, University of Massachusetts, Amherst, MA, USA. [2]Department of Microbiology, University of Massachusetts, Amherst, MA, USA. [3]Institute for Applied Life Sciences (IALS), University of Massachusetts, Amherst, MA, USA. [4]Department of Biomedical Engineering, University of Massachusetts, Amherst, MA, USA. ✉e-mail: dlovley@umass.edu; juny@umass.edu

increased material cost and lower stability (<12 h) reduced practicality. The use of biomaterial such as wood exploits its innate porous structure and built-in hygroscopic groups for ease in fabrication[12], with the benefit of reducing production wastes associated with conventional inorganic materials. However, the energy density remains limited to nW/cm² level, and the rigid and bulk form further limits scalable integration and wearable implementation.

Microorganisms are ubiquitous in nature[13]. Diverse electroactive microorganisms are capable of generating electricity from organic matter oxidation and such 'microbial fuel cells' might be employed for powering microelectronics[3,14,15]. However, the requirement of maintaining cell viability with continuous feedstocks and suitable conditions has limited their practical applications.

Here, we show that biofilms produced from sustainable feedstocks can function as a nonliving biomaterial for evaporation-based electricity production. Current production scales directly with biofilm-sheet size and skin-patch devices harvest sufficient electricity from the moisture on the skin to continuously power wearable devices. The results demonstrate that appropriately engineered biofilms can outperform engineered materials without the need for further processing. The energy output is comparable to that achieved with similar-sized biofilms catalyzing current production in microbial fuel cells[14,15], without the need for an organic feedstock or maintaining cell viability. The ubiquity of biofilms in nature[13] suggests the possibility of additional sources of biomaterial for evaporation-based electricity generation and the possibility of harvesting electricity from diverse aqueous environments.

## Results

### Basic device and performance

*Geobacter sulfurreducens* strain CL-1 is a genetically modified strain that produces highly cohesive, electrically conductive biofilm

sheets (<100 μm thick) that could potentially serve as a sustainably produced, functional biopolymer[16]. In order to determine whether the biofilm sheets were capable of the evaporation-based current generation, the material was produced as previously described[16] (Supplementary Fig. S1) yielding cohesive biofilm sheets ~40 μm thick (Supplementary Fig. S2). The sheets (Fig. 1a) were patterned for device fabrication with standard thin-film laser writing (Fig. 1b; Supplementary Fig. S3). Transmission electron microscopy (TEM) confirmed cells within an extracellular polymeric matrix[17] with nanofluidic channels (width, 100–500 nm) between the cells (Fig. 1c), as previously reported[16].

Electricity harvesting from water evaporation was initially evaluated in a device fabricated by placing two gold electrodes across a biofilm supported on a glass substrate (Fig. 1bi; Supplementary Fig. S4). Immersing one terminal in water yielded a spontaneous voltage output of ~0.45 V, which was sustained for more than a month (Supplementary Fig. S4). The electric outputs were the same when the inert gold electrodes were replaced with carbon electrodes (Supplementary Fig. S5). Depleting the water source depleted the energy output (Supplementary Fig. S6). The voltage output declined with a decrease in the electrode spacing (Supplementary Fig. S7). These results are consistent with an evaporation-based streaming mechanism[8,9]. The device maintained a steady current output (~1.5 μA) for over 30 days (Fig. 1d). The estimated power density generated (~1 μW/cm²) was more than an order of magnitude larger than that previously achieved with carbon and wood materials[10,12,18]. The evaporation-based energy production was comparable to that achieved with similar-sized biofilms catalyzing current production in microbial fuel cells[14,15], without the need for continuously providing an organic feedstock or maintaining cell viability. The energy output increased linearly when the biofilm-sheet devices were connected in arrays (Fig. 1bii; Supplementary Fig. S8), sufficient to power up electronics (Fig. 1e).

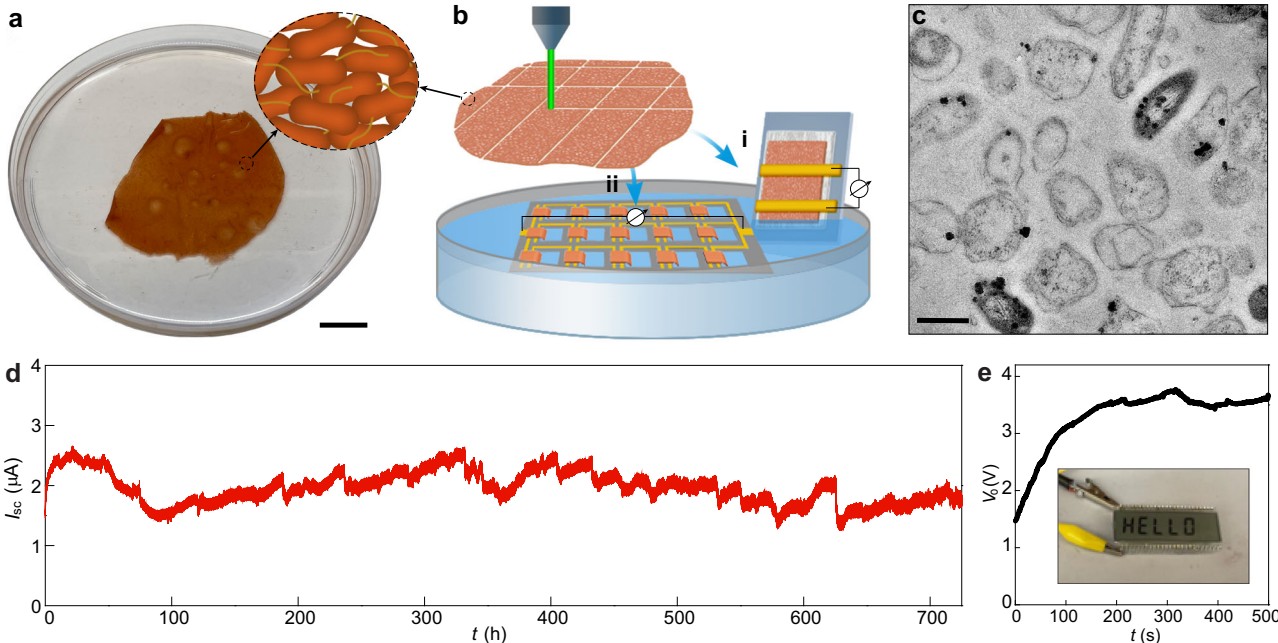

**Fig. 1 | Electric outputs from *G. sulfurreducens* biofilms. a** A harvested biofilm sheet of *G. sulfurreducens* strain CL-1 (schematic inset), floating on water. Scale bar, 2 cm. **b** Schematic of using laser-patterned biofilms to construct (i) single device and (ii) interconnected device array on a PDMS substrate (gray), with a portion of the biofilm at one electrode immersed in water. A tissue paper was used to support the biofilm in the single device, which may assist water evaporation but does not contribute to electric signals (supplementary Fig. S4). **c** Cross-sectional TEM of a *G. sulfurreducens* strain CL-1 biofilm. Scale bar, 1 μm.

**d** A continuous recording of the short-circuit current ($I_{sc}$) from a device for one month. The device had the structure shown in (b-i), with an electrode spacing 2 mm and lateral width 1 cm, yielding an estimated energy density $V_o·I_{sc}/4$ of ~1 μW/cm² in the active biofilm region. The test was performed in the ambient environment with relative humidity (RH) fluctuating between 30–55%. **e** Open-circuit voltage ($V_o$) from an integrated device array (**b**ii) floating on a water surface, which was used to power up an LCD (inset). The actual device is shown in supplementary Fig. S8.

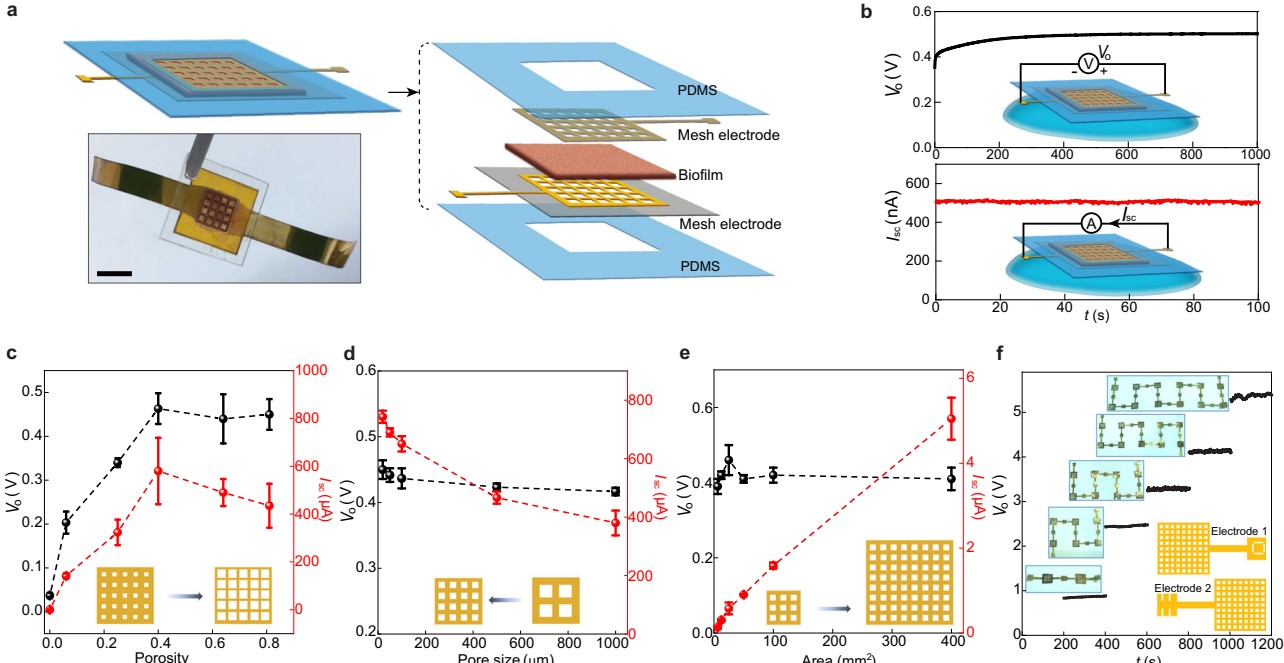

**Fig. 2 | Integrated biofilm devices using mesh electrodes. a** Schematic and actual photo (bottom) of a biofilm device. Scale bar, 0.5 mm. **b** Representative open-circuit voltage $V_o$ (top) and short-circuit current $I_{sc}$ (bottom) from a device placed on a water surface (schematics). The device size was 5 × 5 mm². **c** $V_o$ (black) and $I_{sc}$ (red) from devices with respect to different porosities in the mesh electrodes. The pore size was kept 100 μm and the device sizes were 25 mm². **d** $V_o$ (black) and $I_{sc}$ (red) from devices with fixed porosity of 0.4 but varying pore sizes (from 1000 to 20 μm) in the mesh electrodes. The device sizes were 25 mm². At the pore size of

20 μm, the estimated optimal energy density ($V_o \cdot I_{sc}/4$) was -0.84 μW/cm² in the active biofilm region. The previous device structure (Fig. 1bi) yielded higher density, probably because the biofilm-tissue paper interface improves water transport. **e** $V_o$ (black) and $I_{sc}$ (red) from devices of different areas, with the porosity and pore size in the mesh electrodes, kept 0.4 and 100 μm. **f** Measured $V_o$ from devices connected in series (upper inset) using a "buckle" design in electrodes (bottom schematic). All the tests were performed in the ambient environment (RH-50%). All the error bars are standard deviations.

## Scalable device integration

A vertical thin-film device was constructed to improve integration for scalable power production. The biofilm was sandwiched between a pair of mesh electrodes[19] (Fig. 2a; Supplementary Fig. S9) and the device was sealed with a pair of polydimethylsiloxane (PDMS) thin layers for structural stability. Placing the device on a water surface resulted in a spontaneous voltage output -0.45 V (Fig. 2b, top), consistent with a vertical streaming potential derived from water evaporation across the film.

The conformal device structure enabled the quantification of the evaporation dynamics across the biofilm to further analyze the streaming effect (Supplementary Fig. S10). Voltage outputs increased with an increase in evaporation rate in a manner consistent with the predicted trend for streaming potential (Supplementary Fig. S11). The typical maximal streaming efficiency that has been obtained from engineered single-channel microfluidic devices is -0.1–1%; higher efficiency is possible but requires an ultrasmall nanochannel and specific ionic solutions[20]. Realistic devices built from porous films have much lower efficiency, probably because the stochastic distribution and convolution of the porous fluidic paths reduce the drag efficiency. The biofilm-sheet device achieved an estimated efficiency -0.02% (Supplementary Fig. S12), approaching the maximal range that has been achieved in engineered single channels. Water evaporation through the biofilms was even faster than from an open water surface (Supplementary Fig. S13), which may account for the rapid establishment of the streaming potential over a short distance in the biofilm plane (Supplementary Fig. S7). A similar trend was maintained across the vertical thickness of biofilms (Supplementary Fig. S14), consistent with an isotropic transport expected from the 3D distributed porosity of the biofilm (Fig. 1c). The enhanced streaming efficiency and evaporation rate in the biofilm can account for the improved energy density

compared to other thin-film devices[10,12,18]. Connecting the two mesh electrodes yielded continuous current output (Fig. 2b, bottom).

Due to the quick establishment of streaming potential over a short distance in the biofilm, the voltage output was largely independent of film thickness and a single layer could attain optimal value (Supplementary Fig. S14). The structure of the mesh electrode was further adjusted for optimal energy output. Increasing the porosity of the electrodes increased the voltage and current output up to porosity of 0.4, with no additional benefit from greater porosity increases (Fig. 2c). The saturation trend in voltage is consistent with the concept that the limiting factor in the rate of water transport determining the streaming potential is eventually removed beyond a certain porosity. The current output gradually decreased at porosities above 0.4 (Fig. 2c, red), indicating that upon voltage saturation any further increase in porosity reduces the contact area for charge collection. With a fixed porosity of 0.4, reducing the pore size or increasing the grid density, which is expected to reduce the overall sheet resistance in the device, further improved current collection (Fig. 2d).

We used a pore size of 100 μm that could be conveniently defined by laser writing to study the current production during scalable device integration. There was a linear increase in current output with an increase in device size, while the voltage remained constant (Fig. 2e). The results indicated that the power output can be scaled up in biofilm-sheet devices integrated with mesh electrodes. Device modules were easily connected for a tunable voltage output with a 'buckle' design that provided a rapid connect/disconnect (insets, Fig. 2f; Supplementary Fig. S15). The output voltage linearly increased with an increase in the number of devices, with a voltage of -5 V readily obtained by connecting 12 devices in series (Fig. 2f). The biofilm-sheet arrays can be also integrated on the same flexible substrate to increase the output (Supplementary Fig. S16).

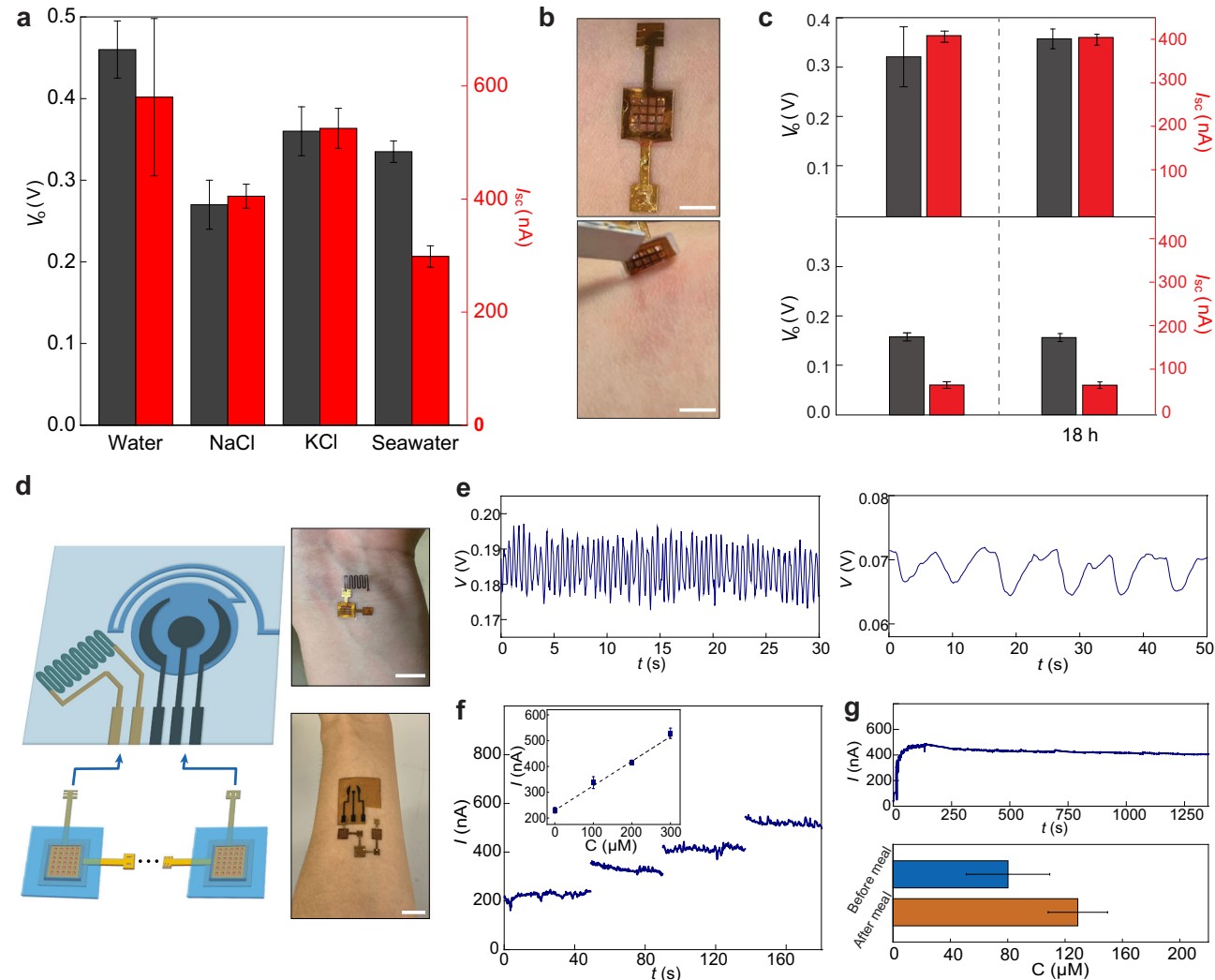

**Fig. 3 | Wearable powering. a** Open-circuit voltage $V_o$ (gray) and short-circuit current $I_{sc}$ (red) from biofilm devices placed on deionized water, 0.5 M NaCl, 0.5 M KCl, and artificial seawater (475.7 mM NaCl, 10.8 mM CaCl$_2$, 25.6 mM MgCl$_2$·6H$_2$O, 28.2 mM MgSO$_4$) solutions. **b** A biofilm device was patched on the skin (top) and removed 18 h later (bottom). Scale bars, 0.5 cm. **c** $V_o$ (gray) and $I_{sc}$ (red) from biofilm devices patched on sweating skin (top) and dry skin (bottom), before (left) and after (right) 18 h. **d** (Left) schematic of connecting biofilm devices to wearable sensors for wearable powering. (Right) Actual photos of powering a skin-wearable strain sensor with one biofilm device (top) and an electrochemical glucose sensor with three biofilm devices (bottom). Scale bars, 1 cm. **e** Measured pulse signal (left) from the wrist and respiration signal (right) from the chest using the biofilm-powered strain sensor. **f** Amperometric responses from a biofilm-powered glucose sensor placed in solutions having glucose concentrations ($C$) of 0, 100, 200, and 300 μM, respectively. The inset shows the calibrated response curve. **g** (Top) A continuous measurement of current from a biofilm-powered glucose sensor during exercise. (Bottom) Calibrated glucose levels from collected measurements before (blue) and after (orange) a meal. All the error bars are standard deviations.

## Powering wearable devices

High salt concentrations diminish the electric output (e.g., <5% of value in water) of typical evaporation-based current generation devices[10,12,18] because the decreased Debye length at high ionic strength reduces the double-layer overlap and hence the streaming efficiency[20]. However, the biofilm-sheet devices maintained electric outputs (e.g., >50% of values in water) in salt solutions and seawater with an ionic concentration of 0.5 M. (Fig. 3a). This can be attributed to the special material properties of the biofilm. Specifically, unlike many inorganic materials, which have monolithic surface charge state, the biofilm contains many amphiphilic groups[21]. The resultant amphiphilic surface may help to effectively repel both positive and negative ions. As a result, the local ionic strength is reduced or the effective Debye length is increased. This effect was observed in other porous organic materials[22,23]. Furthermore, the biofilm has a high density of protein nanowires synthesized by *G. sulfurreducens* for charge transport[6]. These ultrasmall nanowires (e.g., 3 nm diameter) do not block water

transport but introduce a high-density network of water-solid interfaces to effectively increase the number of double layers (Supplementary Fig. S17). Together, the increases in both the effective Debye length and the number of double layers can help to maintain the double-layer overlap or streaming efficiency at high ionic strength. These results show that the biofilm-sheet devices can be used in diverse aqueous environments for energy harvesting.

Since the biofilm-sheet devices maintained the electric outputs in ionic strength higher than that (~0.15 M) in bodily fluid, they were employed for powering skin-wearable devices. The device with the PDMS seal (Fig. 2a) facilitated skin adhesion and the porous biofilm provided a breathable interface, so that prolonged (>12 h) contact did not irritate the skin (Fig. 3b). A device patch worn on sweaty skin produced power comparable to that produced with the salt solution and maintained its performance after 18 h (Fig. 3c, top). Even non-sweating skin generated a substantial electric output (Fig. 3c, bottom), demonstrating that a continuous low-level secretion of moisture from

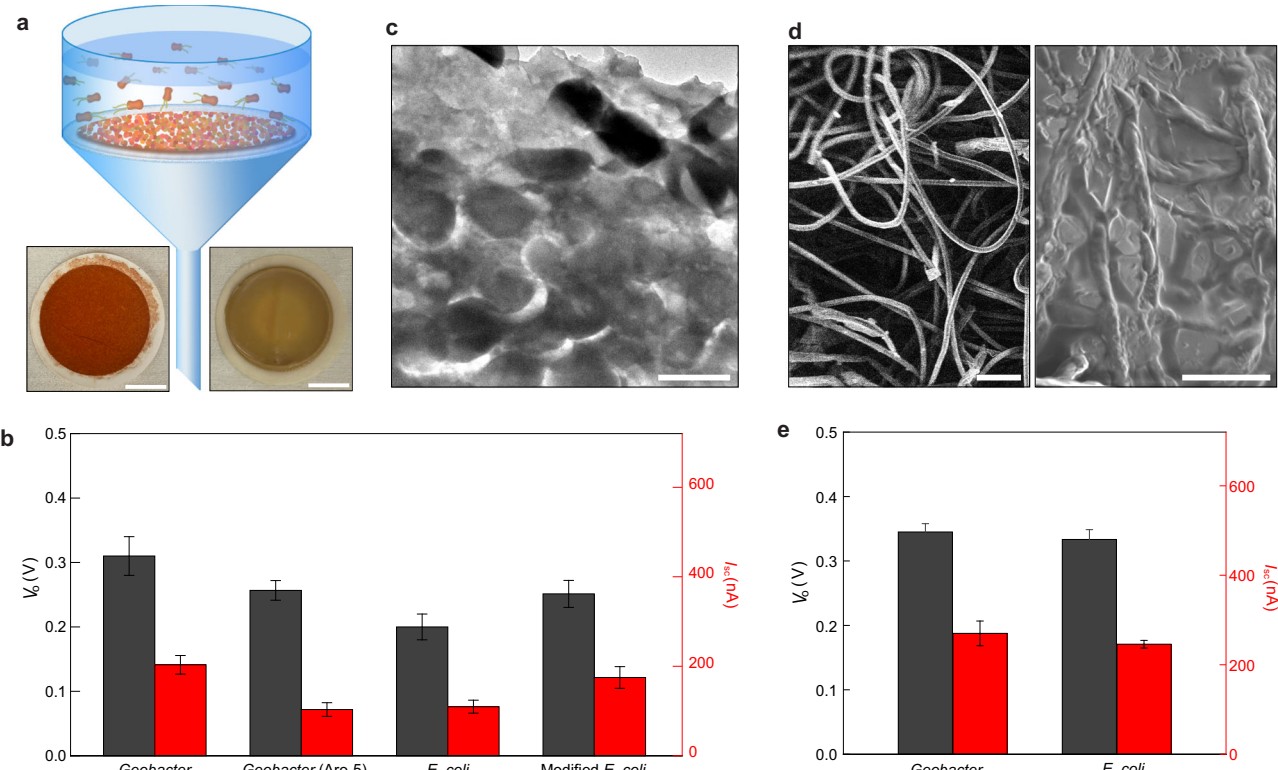

**Fig. 4 | Devices made from filtered biofilm-mats. a** Schematic of harvesting biofilm-mats by filtering microbial solutions. The bottom photos show filtered (left) *G. sulfurreducens* and (right) *E. coli* mats, respectively. Scale bars, 1 cm. **b** Average $V_o$ (gray) and $I_{sc}$ (red) measured from devices fabricated with biofilm-mats of *G. sulfurreducens*, genetically modified *G. sulfurreducens* Aro-5 strain, *E. coli*, and genetically modified *E. coli* strain. The devices had the same size of $5 \times 5$ mm², with a porosity of 0.4 and pore size of 100 μm in the mesh electrodes. **c** Cross-sectional TEM image of a biofilm-mat assembled by filtered *E. coli*. Scale bar, 1 μm. **d** Scanning electron microscope images of (left) a tissue paper and (right) a tissue paper infiltrated with *E. coli*. Scale bars, (left) 100 μm, (right) 10 μm. **e** Average $V_o$ (gray) and $I_{sc}$ (red) measured from devices fabricated with tissue paper infiltrated with *Geobacter* and *E. coli*. The devices had the same size of $5 \times 5$ mm², with a porosity of 0.4 and pore size of 100 μm in the mesh electrodes. All the error bars are standard deviations.

the skin is sufficient to drive this hydroelectric output. Thus, the biofilm-sheet device, complementary to protein nanowire humidity generators[24–26], is a promising candidate for the continuous powering of wearable electronics.

As a proof-of-concept demonstration, skin-wearable biofilm-sheet devices were connected to wearable sensors (Fig. 3d). These included a strain sensor for self-powered monitoring of pulse (Fig. 3e, left), respiration (Fig. 3e, right), and other bodily signals (Supplementary Fig. S18). Interconnected biofilm-sheet devices were able to power a laser-patterned[27,28] wearable electrochemical glucose sensor (Fig. 3f; Supplementary Fig. S19) that monitored the glucose in sweat during exercise (Fig. 3g, top panel), and differentiated glucose levels before and after eating (Fig. 3g, bottom panel).

**Current generation with different biofilms**

The previously recognized mechanism for electricity generation with biofilms of electroactive microbes like *G. sulfurreducens* is their ability to oxidize organic substrates with electron transfer to electrodes in microbial fuel cells[3]. However, this technology requires that the microbes be maintained in a viable state under anoxic conditions with a continuous supply of organic fuel. In contrast, the biofilm-sheet devices achieved comparable or better energy densities and a faster start-up time[14] while not dependent on cell viability. Storing the biofilms in the air for more than a month or baking them at 90 °C had no impact on the current generation (Supplementary Fig. S4c; Supplementary Fig. S20).

The reason that the biofilms-sheet devices generate more power than wood-based systems[12] for evaporation-based electricity generation is probably related to the high surface area of hygroscopic and charged moieties within the biofilm[17], coupled with abundant channels for fluid flow, that are associated with the spacing of small microbial cells in the biofilm matrix (Fig. 1c). These features enhance both water and charge transport. The estimated surface area within the biofilm (~8 m²/cm³) is much greater than that of wood[12]. Biofilm-sheet devices still generate more power than films assembled from nanomaterials of smaller sizes[10,18,29], because these nanomaterials lack a supportive matrix and hence have tight physical contact between individual nanoelements.

An alternative approach to growing biofilms may be to collect planktonic cells as a mat on a filter. This approach was previously explored for making films of *G. sulfurreducens* protein nanowires for electricity production from atmospheric humidity[30]. Devices for evaporation-based power generation fabricated with mats of filtered *G. sulfurreducens* (Fig. 4a; Fig. S21) generated an average voltage of 0.31 V and a current of 0.20 μA (Fig. 4b), or 67% and 35% values of the same size biofilm-sheet devices (Fig. 3a). Devices fabricated with mats of genetically modified *G. sulfurreducens* strain Aro-5 that expresses low-conduction protein nanowires (e.g., 1000-fold lower than wildtype)[31], generated an average voltage of 0.27 V and a current of 0.10 μA (Fig. 4b), or 59% and 18% values of the same size biofilm-sheet devices. To determine if evaporation-based electricity generation is possible with mats of other microbes, mats were fabricated with *Escherichia coli*, a microbe amenable to mass cultivation. Devices fabricated with *E. coli* mats generated an average voltage of 0.20 V and a current of 0.11 μA (Fig. 4b), or 43% and 19% values of the same size biofilm-sheet devices. Devices fabricated with mats of genetically modified *E. coli* expressing conductive protein nanowires[32], generated an average voltage of 0.25 V

and a current of 0.18 μA (Fig. 4b), or 54% and 31% values of the same size biofilm-sheet devices. These results demonstrate generic evaporation-based energy production from diverse biofilms, which are not dependent on cell viability (Supplementary Fig. S4c; Supplementary Fig. S20) and indicate the presence of conductive nanowires emanating from cells is not a major contributor to the current generation. Mats of different microbial species produced similar electric outputs, further suggesting that material conductivity does not directly affect energy production, which is consistent with the general description of the streaming mechanism[20]. Meanwhile, all the biofilm mats had electric outputs lower than the biofilm sheets, suggesting that the microstructure within the films plays an important role in determining energy efficiency. Transmission electron microscopy revealed that, due to the lack of an extracellular polymeric matrix, the porosity of the filtered mats (Fig. 4c) was much lower than in the biofilm sheets (Fig. 1c), which is expected to reduce fluid and charge transport.

These results suggested that improving the microstructure of filtered biofilm mats by adding a supportive scaffold might increase electricity generation. As a proof-of-concept demonstration, planktonic cells were infiltrated within a tissue paper with a microporous structure (Fig. 4d). Devices fabricated from these hybrid mats of *G. sulfurreducens* generated an average voltage of 0. 35 V and a current of 0.27 μA (Fig. 4e), or 113% and 129% produced by the same size devices fabricated solely from mats of *G. sulfurreducens*. Devices fabricated from hybrid mats of *E. coli* generated an average voltage of 0. 33 V and a current of 0.25 μA (Fig. 4e), or 165% and 227% produced by the same size devices fabricated solely from mats of *E. coli*. These results indicate that engineering biofilm composites can improve evaporation-based energy production. Future efforts in composite engineering may include directly culturing planktonic cells in a scaffold sheet or filtering solutions of mixed nanofibers and cells.

Further rational optimization of microbial films for electricity generation will need a better understanding of how the streaming current is generated, which currently is poorly understood[9]. Based on our experimental evidence, a model that assumes the induction of image electrons in the materials can account for the closed-loop current flow without the involvement of the redox process (Supplementary Fig. S22).

## Discussion
Our results demonstrate that biofilm sheets are an innovative, sustainably produced material capable of scalable power production from evaporation-based electricity generation. Other strategies for organizing microbial cells into highly channelized, high surface area materials may be feasible. The ubiquity of microorganisms and their proclivity for biofilm formation suggests possibilities for harvesting electricity via similar evaporation-based strategies in diverse environments.

## Methods
### Biofilm preparation
The conductive biofilms of *G. sulfurreducens* strains CL-1 were routinely cultured as previously described[16]. Briefly, the biofilms were grown anaerobically in two-chambered H-cell systems with a continuous flow of medium with acetate (10 mM) as the electron donor and a polished graphite stick anode poised at 300 mV versus Ag/AgCl as the electron acceptor. For studies with mats of cells collected with filters, wild-type *G. sulfurreducens* was grown as previously described[33] under anaerobic conditions with acetate as the electron donor and fumarate as the electron acceptor. Planktonic cells of *E. coli* were grown in LB medium as previously described[34]. Genetically edited *G. sulfurreducens* Aro-5 strain and *E. coli* strain

were grown as previously described[31,32]. Planktonic cells were filtered through a filter paper (42.5 mm dia., 8 μm pore size; Whatman) (Supplementary Fig. S21). The filtered mat of cells was then washed with deionized (DI) water.

### Fabrication of biofilm devices
The biofilms were patterned using a laser writer (Supplementary Fig. S3) and subsequently transferred onto substrates for device fabrication. Fabrication details for the planar single device (Fig. 1bi) and array device (Fig. 1bii) can be found in Supplementary Fig. S4 and Supplementary Fig. S8, respectively. The mesh electrode was fabricated by coating a thin gold layer on a laser-patterned polyimide (PI) mesh scaffold (Supplementary Fig. S9).

### Imaging
In order to examine cross sections of biofilms with TEM biofilms were fixed (2% paraformaldehyde and 0.5% glutaraldehyde in 50 mM 1, 4-piperazinebis (ethanesulfonic acid) (PIPES) at pH 7.2) for 1 h at room temperature and washed 3 times with 50 mM PIPES. The biofilm samples were then dehydrated with a graded ethanol series (30, 50, 70, 80, 100%; 30 min each stage with gentle agitation). The dehydrated samples were infiltrated with LR White (medium grade, Electron Microscopy Sciences) and polymerized at 55 °C overnight. Thin sections (~50 nm thick) of fixed biofilm were made with a microtome (ULtracut S; Leica Microsystems). Sections were positively stained with 2% uranyl acetate and imaged (FEI Tecnai-T12 TEM) at 80 KV. The thickness of biofilms was acquired using 3D profiler (NewView™ 9000; Zygo).

### Fabrication of glucose sensors
The carbon electrodes were defined by a laser writer (LaserPro Spirit GLS; GCC) on a polyimide substrate[27,28]. An Ag layer was electrodeposited onto the reference electrode by immersing it in a plating solution (250 mM silver nitrate, 750 mM sodium thiosulfate, and 500 mM sodium bisulfite), applied with −0.2 mA for 100 s. The rest electrodes were electrochemically deposited with a Pt layer, by immersing them in a solution (1.0 mM $H_2PtCl_6$ dissolved in 0.5 M $H_2SO_4$) and scanning the electrodes from +0.5 to −0.7 V for consecutive 30 cycles at 100 mV/s. The glucose oxidase (GOx) enzyme was subsequently immobilized by dip-coating the Pt electrode in a solution containing 140 mg/ml GOx, 56 mg/ml bovine serum albumin, and 25 w % glutaraldehyde, followed by a 2-h soak in phosphate-buffered saline to remove residue. A double-sided medical adhesive (MH-90445Q; Adhesives Research), defined with a microfluidic channel (200 μm wide) and reservoir (5 mm diameter) by the laser writer, was attached to the PI substrate with defined electrodes. The sensing signals were acquired using an electrochemical workstation (CHI 440; CH Instruments).

### Fabrication of strain sensors
The serpentine electrode (0.5 mm wide, 1 cm long, Cr/Pt = 10/20 nm) was defined on a polyimide film by standard photolithography, metal evaporation, and lift-off processes. Cracks were generated in the electrode by stretching the substrate (2%) using a mechanical testing system (ESM303; Mark-10 Inc.). The sensor was attached to the skin with a double-sided medical adhesive.

### Electrical measurements
Electrical measurements were performed in the ambient environment unless otherwise specified. The biofilm devices in Fig. 1bi were tested by immersing one end of the biofilm into a petri dish filled with DI water. The biofilm devices with mesh electrodes (Fig. 2a) were tested by placing them on wet tissue papers soaked with DI water. The voltage and current outputs were measured by using a Keithley 2401.

## Data availability

All data needed to evaluate the conclusions in the paper are present in the main text and the supplementary materials. Additional data related to this paper may be requested from the authors upon reasonable request.

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

## Acknowledgements

J.Y. and D.R.L. acknowledge support from the National Science Foundation (NSF) DMR2027102. X.L. acknowledges support from the Link Foundation Energy Fellowship. J.Y. also acknowledges support from NSF CAREER CBET-1844904, NSF ECCS-1917630, and ONR N00014-21-1-2593. Part of the device fabrication work was conducted in the clean room of the Center for Hierarchical Manufacturing (CHM), an NSF Nanoscale Science and Engineering Center (NSEC) located at the University of Massachusetts Amherst.

## Author contributions

J.Y. and X.L. conceived the project and designed experiments. D.R.L. oversaw material design and production. X.L. carried out experimental studies in material characterization, device fabrication, and electrical measurement. H.G., T.F., S.F., and L.S. helped with device fabrication and characterization. T.U., T.L.W., and K.P.N. prepared biofilms and bacteria solution. J.Y., D.R.L., and X.L. wrote the paper. All authors discussed the results and implications and commented on the manuscript.

## Competing interests

The authors declare no competing interests.
