## [Peer Review File · Nature Communications]

Microbial Biofilms for Electricity Generation from Water Evaporation and Power to WearablesREVIEWER COMMENTS

Reviewer #1 (Remarks to the Author):

In this manuscript, the authors report a general strategy for evaporation-based electricity generation using biofilm as the platform material. Sustainable electric outputs from cultured *G. sulfurreducens* biofilms have been recorded, and the effects of biofilm geometry (through laser patterning), electrode configuration, and ionic concentration are carefully examined. The author also compared the performance between naturally grown biofilms and artificially filtered cell layers, and demonstrate the importance of extracellular polymeric matrix in the process. Lastly, the authors present proof-of-concept demonstration for powering a range of skin wearable devices. Overall, the current work presents an interesting investigation of the biofilm-based electricity generation through water evaporation, which could open up new possibilities for powering skin-wearable devices and many other electronics. Publication in Nature communications is highly recommended; meanwhile, it would also be helpful to address the following questions in the revised manuscript to better illustrate the underlying mechanism for future device design/optimization.

1. Thermodynamically, what's the energy conversion efficiency that the device can achieve? And what could be the physical limits/factors affecting the device performance?
2. How does biofilm conductivity affect power generation (e.g. between different *Geobacter* strains, or different bacteria species)? And cell viability?
3. It seems the biofilm microstructure plays important roles in the energy transfer process. It could be interesting to explore different physical/chemical/biological tools to further investigate/optimize the biofilm structure for future device design.

Reviewer #2 (Remarks to the Author):

The manuscript from Liu and colleagues presents an energy harvesting system using microbial biofilms. They demonstrate wearable sensors powered by these energy harvesters. The power outputs of centimeter scale devices are around 1 microwatt, which small, yet sufficient for practical uses. The power output appears to be larger than devices that seem to be functioning under similar conditions. In that respect, I find the demonstrations and the general concept quite interesting. However, I am not convinced that the mechanism of the electrical power generation is the streaming currents. The authors present their results in this context and highlight results that appear to be consistent with an interpretation based on streaming currents. Therefore, I would like to expand my critique on this issue.

1. Line 65: "Depleting the water source depleted the energy output (fig. S6)". On its own, this observation suggests that the presence of water is necessary for power generation. However, it is difficult to use this observation to argue that power is generated from streaming currents. The authors correctly add a second test and refer to "increasing the evaporation rate led to an increase in energy output (Fig. S7)." The underlying idea of this test, the relationship between water flow rate and the open circuit voltage, offers a quantitative test to support their claim, but the data in Fig. S7 is not analyzed quantitatively and in connection with theoretical modelling of streaming currents. From what I see, when the temperature changes from 25 to 40 degrees Celsius, the open circuit voltage increases from about 0.44 V to about 0.53 V (note also that the error bar wasn't defined in this figure). This appears to be inconsistent with a streaming mechanism. If the evaporation rate is proportional to the vapor pressure of water, then one would expect the open circuit voltage to increase substantially, by a factor of 3 or so (The vapor pressure of water doubles with every 10 degrees Celsius). The short circuit current should increase similarly. If a quantitative argument is not

given, a qualitative reasoning is enough to make a claim because most mechanisms that lead to generation of electrical potentials have some temperature dependency. There isn't enough discriminatory power of a qualitative argument in this particular case.

2. Lines 76-79: "The biofilm established the streaming potential over a short distance along both the vertical thickness and in the planar direction (Fig. S10), consistent with an isotropic transport expected from the 3D distributed porosity in the biofilm (Fig. 1c)." While the authors use this reasoning to argue that the observation is consistent with a streaming mechanism, it could also be a counterargument. Both demonstrations yield similar open circuit voltages. This could possibly be the case if water pressure drop across the electrodes is constant in both scenarios, but in the experiment of Fig. S10, the electrode spacing is varied significantly while the biofilm configuration is kept the same (at least this is what I am able to interpret from the illustration) and that suggests the pressure drop would be different among different electrode spacings. A key piece of information could have been gleaned from the variation of open circuit current with electrode spacing, but there is only one set of measurements (corresponding to the shortest electrode spacing), so a comparison cannot be made. A comparison with a theoretical modeling of streaming currents could have been informative and perhaps conclusive.

3. Streaming currents are supposed to be carried by migrating ions, but the external circuit uses electrons. So, at some point these ions are transferring their charges to the electrodes. One would expect some change in the biofilm or gas formation depending on the ions involved, especially after month-long current measurements. Is there such evidence?

4. Line 92-95: "High salt concentrations diminish the power output of some evaporation-based current generation devices (13) because the decreased Debye length at high ionic strength reduces the double-layer overlap and hence the streaming efficiency (19). However, the biofilm-sheet devices maintained electric outputs in salt solutions with an ionic concentration (0.5 M) higher than that (~0.15 M) in bodily water (Fig. 3a)." This is a good result that demonstrates the capability of their device, but it goes strongly against the claim that the power generation is due to streaming. I recommend authors inspect Figures 2 and 3 of reference 19 that they cite here. If the open circuit voltage and current do not change by orders magnitude at 0.5 M NaCl or KCl (The data in Fig. 3a suggest they do not), then the authors should look for an alternative explanation. These results are hard to reconcile with a streaming current mechanism.

I believe it would be unfair to the authors to ask them to find out the mechanism of the power generation in one publication. This appears to be a complex problem. The primary result that they are able generate current with an open circuit voltage around 0.4 V is quite interesting and it could stimulate further research towards its origin. However, I would object claims regarding the mechanism without adequate evidence. For example, their claim in the title "generating electricity from water evaporation through microbial biofilms" is not adequately supported in this reviewer's opinion (as explained above in items 1-4). I agree that water is necessary and electricity is being generated, but the data seems to be going against a streaming current mechanism. Could there possibly be another mechanism that is facilitated by evaporation?

Minor comments:

- i. Line 317: the sentence ending with planktonic seems to be incomplete.
- ii. Fig. S6: It's not clear to me what "depletion of water source" means. Do the authors stop water supply and then the level of water gradually goes down? If so, why would the voltage gradually go down? The drop in voltage seems to have an exponentially decaying characteristic. This data could be informative if analyzed quantitatively.
- iii. Fig. 1a, the schematic on the right side of the panel: What is the grey square around the bottom mesh electrode?
- iv. Error bars should be defined in all figure panels that present error bars.

Reviewer #3 (Remarks to the Author):

The work demonstrated a creative study showing the benefits of using microbial biomaterials for electricity harvesting from water evaporation (harnessing the 'streaming current' mechanism). Several highlights of the work include:

1. Bringing in a concept in harnessing microbial materials for fabricating 'hydrovoltaic' devices. The revealed generic effect shows the potential to broadly explore this renewable material source for clean energy, with expected benefits in ease of material production, abundance, environmental friendliness.
2. Achieving exceptional energy density and current sustainability compared to devices/technologies fabricated from traditional engineered materials.
3. The material composition and structure in the microbial film overcome the energy decay (resultant from Debye screening) in salty water observed in conventional materials. It has the implication to push forward the field of microbattery or micro fuel cell (both requiring chemical feed).

Given above reasons, I highly recommend this work to be published in Nat. Commun. Some suggestions for potential improvements:

- i. Electrode-grown microbial biofilms show better energy efficiency than biofilms made from filtering, due to the collapse of inter-cellular space during filtering. Given that solution/filtering processing may still have the advantage in higher production throughput, is there any way to engineer improved porosity in filtered biofilm to increase energy density?
- ii. The scalability of the device has been well studied in terms of device planar size. What is the performance/scalability with respect to the film vertical thickness? Will the pore size in the electrode still have considerable effect on the results?
- iii. It would be useful to see if the devices can work with sea water for broader potential.

Response to Reviewers

Reviewer #1:

*In this manuscript, the authors report a general strategy for evaporation-based electricity generation using biofilm as the platform material. Sustainable electric outputs from cultured *G. sulfurreducens* biofilms have been recorded, and the effects of biofilm geometry (through laser patterning), electrode configuration, and ionic concentration are carefully examined. The author also compared the performance between naturally grown biofilms and artificially filtered cell layers and demonstrate the importance of extracellular polymeric matrix in the process. Lastly, the authors present proof-of-concept demonstration for powering a range of skin wearable devices. Overall, the current work presents an interesting investigation of the biofilm-based electricity generation through water evaporation, which could open up new possibilities for powering skin-wearable devices and many other electronics. Publication in Nature communications is highly recommended; meanwhile, it would also be helpful to address the following questions in the revised manuscript to better illustrate the underlying mechanism for future device design/optimization.*

We thank the reviewer for the positive comments and constructive suggestions for improving our manuscript. We have accordingly conducted additional experiments for the revised manuscript. Please find below our detailed response addressing each suggestion/question the reviewer raised. To make it clear, we have used *italic* fonts for the reviewers' comments, **black fonts for our replies, and **blue** fonts for revisions.**

1. Thermodynamically, what's the energy conversion efficiency that the device can achieve? And what could be the physical limits/factors affecting the device performance?

Based on the general description of the streaming mechanism, the electricity is induced from the kinetic water flow. Therefore, the conversion efficiency is defined as the ratio between the output electric energy and the input kinetic energy.

Specifically, the input kinetic power $P_{kinetic}$ can be calculated as $P_{kinetic} = Q \cdot \Delta p$, where Q is the volume flow rate and Δp the pressure difference across the biofilm (*Nano Lett.* 6, 2232-2237, 2006). The maximal electric energy output can be approximated as $P_{electric} = V_o \cdot I_s / 4$, where V_o and I_s are the measured open-circuit voltage and short-circuit current, respectively. As a result, the conversion efficiency η can be obtained as:

$$\eta = \frac{P_{electric}}{P_{kinetic}} = \frac{V_o \cdot I_s}{4 \cdot Q \cdot \Delta p} \quad (1)$$

We set up following experiment to determine the volume flow rate Q and corresponding V_o, I_s . In the setup, the biofilm device was used to seal an autosampler vial fully filled with water (Fig. R1a). The vial was placed on a digital weight gauge so that the water loss through evaporation across the biofilm could be monitored (Fig. R1b). The vial was covered with a flexible heating pad to control the water temperature. The power in the heating pad was calibrated by measuring the water temperature with a thermal meter. The water evaporation rates at different temperatures were measured in this way (Fig. R2a). Corresponding electric outputs from the biofilm device were measured under the same conditions (Fig. R2b). We used the data collected at a water temperature (T) of 25 °C to show the calculation of conversion efficiency.

At T = 25 °C, the measured V_o, I_s , and water loss rate were 0.4 V, 0.4 μ A, and 6.89 μ g/s, respectively. The weight loss rate corresponded to a volume loss/flow rate ~ 0.3 cm³/s, by considering that the vapor density is ~ 23 g/m³ (T = 25 °C).

The relative humidity (RH) and temperature outside the vial at the vicinity of the biofilm device were measured by a thin-film sensor (SEK-SHT40-AD1B-Sensors; Sensirion) to be $\sim 86\%$ and 22 °C, whereas the RH and temperature inside the vial can be treated as 100% and 25 °C, respectively. Consequently, $\Delta p = p_{100\%RH,25^\circ C} - p_{86\%RH,22^\circ C} = 8.8 \times 10^2$ Pa.

Inputting these obtained values into equation (1) yields an energy conversion efficiency $\eta = 0.02\%$. This efficiency was maintained at different temperatures (Fig. R2c). Note that the maximal efficiency obtained from engineered single-channel fluidic devices was $\sim 0.1\% - 1\%$ (*Nano Lett.* 6, 2232-2237, 2006). Realistic devices made from thin films had much lower efficiency (e.g., estimated to be $< 0.01\%$), probably because the dispersion and convection in the porous channels reduce the drag efficiency. Thus, the biofilm devices achieved an efficiency much higher than previous thin-film devices.

The results also indicate that further improvement in energy efficiency is possible. For example, improving the cohesiveness and alignment of bacteria in the biofilm is expected to reduce the size and dispersion in nanoscale channels, and hence, improve the energy efficiency. Engineering biofilm composites may also provide a promising route for improvement (see response in #3).

Fig. R1. Measurement setup. (a) An autosampler vial was filled with water, (middle) with the opening of the lid (right) covered with a biofilm device. Scale bar, 5 mm. (b) A flexible heater was used to wrap the vial (middle). (Right) the vial was placed on a digital gauge for monitoring the weight change.

Fig. R2. (a) Measured water evaporation rate at different temperatures. (b) Corresponding measured voltage and current outputs from the biofilm device. (c) Calculated energy efficiency.

In the revised manuscript (page 4), we have added following description: “The typical maximal streaming efficiency that has been obtained from engineered single-channel microfluidic devices is $\sim 0.1 - 1\%$; higher efficiency is possible but requires an ultrasmall nanochannel and specific ionic solutions.²⁰ Realistic devices built from porous films have much lower efficiency, probably because the stochastic distribution and convection of the porous fluidic paths reduce the drag efficiency. The biofilm-sheet device achieved an estimated efficiency $\sim 0.02\%$ (Fig. S12), approaching the maximal range that has been achieved in engineered single channels.” Correspondingly, we have added the experimental details (R1) and measured data and analysis (Fig. R2) as new Supplementary Fig. S11 and S12.

2. How does biofilm conductivity affect power generation (e.g. between different *Geobacter* strains, or different bacteria species)? And cell viability?

According to the general description of the streaming effect induced during water flow through a channel of length L , the streaming potential (i.e., open-circuit voltage) and streaming current

density (i.e., short-circuit current density) are $V_{str} = \frac{\varepsilon_0 \cdot \varepsilon_r \cdot \Delta p \cdot \zeta}{\sigma \cdot \eta}$ and $J_{str} = \frac{\varepsilon_0 \cdot \varepsilon_r \cdot \Delta p \cdot \zeta}{\eta \cdot L}$, respectively, where $\varepsilon_0 \cdot \varepsilon_r$, σ , and η are the permittivity, conductivity, and viscosity of the water solution. ζ and Δp are the zeta potential and pressure difference across the channel. For evaporation-driven flow, the pressure difference can be expressed as $\Delta p = 4 \cdot \gamma \cdot \cos \theta / d$, where γ , θ , d are the surface tension of water, contact angle between water and the capillary channel, and the diameter of the channel (*ACS Appl. Mater. Interfaces* 12, 11232-11239, 2020). Substituting Δp yields:

$$V_{str} = \frac{4 \cdot \varepsilon_0 \cdot \varepsilon_r \cdot \zeta \cdot \cos \theta \cdot \gamma}{\sigma \cdot \eta \cdot d} \quad (2)$$

$$J_{str} = \frac{4 \cdot \varepsilon_0 \cdot \varepsilon_r \cdot \zeta \cdot \cos \theta \cdot \gamma}{\eta \cdot L \cdot d} \quad (3)$$

Consider that a biofilm of cross-section area A contains many of these small channels (defined by a porosity s), then the effective total channel area is $A \cdot s$. The streaming current is then:

$$I_{str} = A \cdot s \cdot J_{str} = \frac{4 \cdot A \cdot s \cdot \varepsilon_0 \cdot \varepsilon_r \cdot \zeta \cdot \cos \theta \cdot \gamma}{\eta \cdot L \cdot d} \quad (4)$$

Above equations (2) and (4) show that the electric outputs are not directly related to material conductivity (note that σ is conductivity of water). Instead, they are related to 1) material surface property that determines ζ and θ (e.g., hydrophilicity), and 2) material microstructure that determines the porosity s and pore size d .

The analysis leads to the general expectations that 1) biofilms of different conductivities can still have similar electric output if they share similar surface and structural properties, and 2) cell viability does not affect the result if it does significantly alter surface and structural properties in the biofilm. To support the conclusions, we have performed additional experiments as follows.

First, we fabricated devices from a genetically modified *Geobacter* strain (Aro-5), which expresses poorly conductive protein nanowires (e.g., 1000-fold lower than wildtype) and is much less electroactive. Since the strain does not grow cohesive biofilm, we used the filtering method to prepare biofilm mats. Devices fabricated from mats of *Geobacter* Aro-5 showed ~50%

Fig. R3. Average V_o (gray) and I_{sc} (red) measured from devices fabricated with mats of filtered *G. sulfurreducens*, genetically modified *G. sulfurreducens* Aro-5 strain, *E. coli*, genetically modified *E. coli* strain, and (right) cohesive-grown *Geobacter* CL-1 strain. The devices had the same size of 5×5 mm².

reduction in electric current production compared to devices made from mats of wildtype *Geobacter* (Fig. R3). Second, devices fabricated from mats of genetically modified *E. coli* expressing conductive protein nanowires (*ACS Synth. Biol.* 9, 647–654, 2020) showed ~30% increase in current production compared to devices made from mats of wild-type *E. coli* not expressing the wires (Fig. R3). These changes in the current production are considered more related to changes in surface/microstructure properties associated with the genetic editing, because the cohesive-grown *Geobacter* strain CL-1 expressing nanowires of same conductivity had much improved (e.g., > 3-fold) current production compared to filtered biofilm mats (Fig. R3, right).

Second, devices made from freshly prepared biofilm-sheets, in which the majority of the cells were still alive, had electric outputs close to devices made from biofilms that were stored for over 30 days or baked at 90 °C (Fig. R4).

Fig. R4. (Left) A 35-day recording of the V_o from a biofilm device. (Right) V_o from a biofilm device before (light blue) and after (dark blue) being heated to 90°C for 30 min.

In the revised manuscript, we have added following related discussion (page 7): “Devices for evaporation-based power generation fabricated with mats of filtered *G. sulfurreducens* (Fig. 4a; Figs. S21) generated an average voltage of 0.31 V and a current of 0.20 μ A (Fig. 4b), or 67% and 35% values of the same size biofilm-sheet devices (Fig. 3a). Devices fabricated with mats of genetically modified *G. sulfurreducens* strain Aro-5 that expresses low-conduction protein nanowires (e.g., 1000-fold lower than wildtype),³¹ generated an average voltage of 0.27 V and a current of 0.10 μ A (Fig. 4b), or 59% and 18% values of the same size biofilm-sheet devices. To determine if evaporation-based electricity generation is possible with mats of other microbes, mats were fabricated with *Escherichia coli*, a microbe amenable to mass cultivation. Devices fabricated with *E. coli* mats generated an average voltage of 0.20 V and a current of 0.11 μ A (Fig. 4b), or 43% and 19% values of the same size biofilm-sheet devices. Devices fabricated with mats of genetically modified *E. coli* expressing conductive protein nanowires,³² generated an average voltage of 0.25 V and a current of 0.18 μ A (Fig. 4b), or 54% and 31% values of the same size biofilm-sheet devices. These results demonstrate generic evaporation-based energy production from diverse biofilms, which are not dependent on cell viability (Fig. S4c; Fig. S20) and indicate that the presence of conductive nanowires emanating from cells are not a major contributor to current generation. Mats of different microbial species produced similar electric outputs, further suggesting that material conductivity does not directly affect energy production, which is consistent with the general description of the streaming mechanism.²⁰ Meanwhile, all the biofilm-mats had electric outputs lower than the biofilm-sheets, suggesting that the microstructure within the films plays an important role in determining the energy efficiency.” We have added Fig. R3 into the new Fig. 4b correspondingly. Fig. R4 was already in Supplementary Figs. 4 & 20.

3. It seems the biofilm microstructure plays important roles in the energy transfer process. It could be interesting to explore different physical/chemical/biological tools to further investigate/optimize the biofilm structure for future device design.

According to the analysis outlined in response #2, microstructure in the biofilm indeed plays an important role for energy efficiency. Specifically, according to equation (4) in response #2, the electric current output is related to the microporosity. We have demonstrated that the biofilm-sheets of *Geobacter* CL-1 strain, which have a high density of 3D porosity due to support from an extracellular polymeric matrix, tripled the current production compared to filtered mats with reduced porosity (Fig. R3). From this result it can be understood that an improved porosity increases the overall surface area for water/charge interaction, and hence, the energy efficiency. This points to a general strategy for improving energy efficiency by modulating the porosity in biofilms, e.g., by introducing an engineered supportive scaffold.

We provided proof-of-concept demonstration by infiltrating bacteria into a tissue-paper scaffold with a 3D porosity to form hybrid mats (Fig. R5a). Both *Geobacter* and *E. coli* were used to show the

Fig. R5. (a) Scanning electron microscope images of (left) a tissue paper and (right) a tissue paper infiltrated with *E. coli*. Scale bars, (left) 100 μm, (right) 10 μm. (b) Average V_o (gray) and I_{sc} (red) measured from devices fabricated with biofilm-mats of *Geobacter*, infiltrated *Geobacter*, *E. coli*, and infiltrated *E. coli*.

generality of the response. The energy outputs (e.g., $\sim V_o \times I_{sc}$) from infiltrated hybrid mats of *Geobacter* and *E. coli* increased $\sim 46\%$ and $\sim 375\%$ compared to values from mats of pure *Geobacter* and *E. coli*, respectively (Fig. R5b).

It is also worthwhile to note that the infiltrated hybrid mats of *Geobacter* and *E. coli* had similar energy outputs despite the difference in cell species, which is consistent with our analysis in response #2.

These results point to the general strategy of improving energy efficiency by engineering hybrid/composite biofilms. Other methods, such as culturing bacteria in a scaffold or mixing nanofibers in bacteria during filtering, can offer broad options for composition engineering. The systematic study of these possibilities is a new direction worthy of future effort.

In the revised manuscript (page 7-8), we have added following description: “These results suggested that improving the microstructure of filtered biofilm-mats by adding a supportive scaffold might increase the electricity generation. As proof-of-concept demonstration, planktonic cells were infiltrated within a tissue paper with microporous structure (Fig. 4d). Devices fabricated from these hybrid mats of *G. sulfurreducens* generated an average voltage of 0.35 V and a current of 0.27 μA (Fig. 4e), or 113% and 129% produced by the same size devices fabricated solely from mats of *G. sulfurreducens*. Devices fabricated from hybrid mats of *E. coli* generated an average voltage of 0.33 V and a current of 0.25 μA (Fig. 4e), or 165% and 227% produced by the same size devices fabricated solely from mats of *E. coli*. These results indicate that engineering biofilm composites can improve evaporation-based energy production. Future efforts in composite engineering may include directly culturing planktonic cells in a scaffold sheet or filtering solutions of mixed nanofibers and cells.” Correspondingly, we have added Fig. R5a and Fig. R5b to a new Figs. 4d, e.

Again, we greatly thank the reviewer for the time, effort and valuable suggestions to help us to improve the work. We believe our efforts have improved the presentation to the quality of publication.

Reviewer #2:

The manuscript from Liu and colleagues presents an energy harvesting system using microbial biofilms. They demonstrate wearable sensors powered by these energy harvesters. The power outputs of centimeter scale devices are around 1 microwatt, which small, yet sufficient for practical uses. The power output appears to be larger than devices that seem to be functioning under similar conditions. In that respect, I find the demonstrations and the general concept quite interesting. However, I am not convinced that the mechanism of the electrical power generation is the streaming currents. The authors present their results in this context and highlight results that appear to be consistent with an interpretation based on streaming currents. Therefore, I would like to expand my critique on this issue.

We thank the reviewer for confirming the value and interest of the work, and also raising critical questions to help improve the quality of the manuscript. Please find below our detailed responses addressing to each of the comments. To make it clear, we have used *italic* fonts for the reviewers' comments, **black fonts for our replies, and **blue** fonts for revisions.**

1. Line 65: “Depleting the water source depleted the energy output (fig. S6)”. On its own, this observation suggests that the presence of water is necessary for power generation. However, it is difficult to use this observation to argue that power is generated from streaming currents. The authors correctly add a second test and refer to “increasing the evaporation rate led to an increase in energy output (Fig. S7).” The underlying idea of this test, the relationship between water flow rate and the open circuit voltage, offers a quantitative test to support their claim, but the data in Fig. S7 is not analyzed quantitatively and in connection with theoretical modelling of streaming currents. From what I see, when the temperature changes from 25 to 40 degrees Celsius, the open circuit voltage increases from about 0.44 V to about 0.53 V (note also that the error bar wasn't defined in this figure). This appears to be inconsistent with a streaming mechanism. If the evaporation rate is proportional to the vapor pressure of water, then one would expect the open circuit voltage to increase substantially, by a factor of 3 or so (The vapor pressure of water doubles with every 10 degrees Celsius). The short circuit current should increase similarly. If a quantitative argument is not given, a qualitative reasoning is not enough to make a claim because most mechanisms that lead to generation of electrical potentials have some temperature dependency. There isn't enough discriminatory power of a qualitative argument in this case.

We thank the reviewer for the thoughtful consideration. To provide more quantitative analysis, we set up the following experimental procedure.

In the setup, the biofilm device was used to seal an autosampler vial fully filled with water (Fig. R1a). The vial was placed on a digital weight gauge so that the water loss through evaporation across the biofilm was monitored (Fig. R1b). The vial was covered with a flexible heating pad to control the water temperature. The power in the heating pad was calibrated by measuring the water temperature with a thermal meter. Corresponding electric outputs from the biofilm device were measured under the same conditions.

Fig. R1. Measurement setup. (a) An autosampler vial was filled with water, (middle) with the opening of the lid (right) covered with a biofilm device. Scale bar, 5 mm. (b) A flexible heater was used to wrap the vial (middle). (Right) the vial was placed on a digital gauge for monitoring the weight change.

The test indeed showed that the rate of water evaporation increased ~100% per 10 °C within the test temperature of 25-40 °C (Fig. R2a). The voltage output also showed a close-to-linearity increase at a rate of ~25% per 10 °C. The fact that this increase in the voltage output is not 100% for each 10 °C V_s , is related to factors described below.

Fig. R2. (a) Measured water evaporation rate at different temperatures. (b) Corresponding measured voltage outputs from the biofilm device. (c) Comparison of (normalized) voltage outputs between measured values (black) and predicted values (red) from the streaming model.

Specifically, the streaming potential in a standard microfluidic channel can be expressed as

$$V_s = \frac{\epsilon_0 \epsilon_r \Delta P \zeta}{\sigma \eta} \quad (1)$$

where $\epsilon_0 \epsilon_r$, σ , η , ζ , and ΔP are the permittivity, conductivity, viscosity of the water solution, material zeta potential, and pressure difference across the channel. If ΔP is the only variable during the process, then it would lead to the expectation of a proportionate increase in V_s .

However, this assumption does not hold because other parameters are also temperature dependent. Temperature-dependent (normalized) values for $\epsilon_0 \epsilon_r$, σ , η in water from the literature (*Phys. Rev.* 35, 623, 1930; *J. Phys. Chem. Ref. Data* 7, 941-948, 1978) (see table below) were incorporated in equation 1. The temperature-corrected prediction from equation 1 shows a trend consistent with our measured values (Fig. R2c).

T (°C)	25	30	35	40
ΔP	1	1.32	1.96	2.52
$\epsilon_0 \epsilon_r$	1	0.97	0.95	0.93
σ	1	1.28	1.63	2.08
η	1	0.90	0.81	0.73

In the revised manuscript (page 4), we have added following description: “The conformal device structure enabled the quantification of the evaporation dynamics across the biofilm to further analyze the streaming effect (Fig. S10). Voltage outputs increased with an increase in evaporation rate in a manner consistent with the predicted trend for streaming potential (Fig. S11).” Correspondingly, we have added the details of experiment (Fig. R1) and data analysis (Fig. R2) in new **Supplementary Fig. S10** and **Fig. S11**.

2. Lines 76-79: “The biofilm established the streaming potential over a short distance along both the vertical thickness and in the planar direction (Fig. S10), consistent with an isotropic transport expected from the 3D distributed porosity in the biofilm (Fig. 1c).” While the authors use this reasoning to argue that the observation is consistent with a streaming mechanism, it could also be a counterargument. Both demonstrations yield similar open circuit voltages. This could possibly be the case if water pressure drop across the electrodes is constant in both scenarios, but in the experiment of Fig. S10, the electrode spacing is varied significantly while the biofilm configuration is kept the same (at least this is what I am able to interpret from the illustration) and that suggests the pressure drop would be different among different electrode spacings. A key piece of information could have been gleaned from the variation of open circuit current with electrode spacing, but there is only one set of measurements (corresponding to the shortest electrode spacing), so a comparison cannot be made. A comparison with a theoretical modeling of streaming currents could have been informative and perhaps conclusive.

It is expected that the streaming potential proportionally increases within the fluidic region but saturates outside this region. Previously, we used laser writing to define electrodes, which yielded relatively large electrode spacings (e.g., $\geq 100 \mu\text{m}$). The spacing may have exceeded the active fluidic region and observed a saturated V_s independent of the electrode spacing.

We have now used photolithography to define electrodes with smaller spacings (e.g., 8, 24, 40, 88 μm). We found that the voltage showed a linear increase with the increase of electrode spacing and gradually saturated beyond 88 μm (Fig. R3). The linear increase was consistent with general streaming description $V_s \sim \Delta p \sim \Delta L \cdot Q$, where ΔL and Q are the fluidic length and flow rate. The results also reveal that the fluidic distance in the biofilm is considerably shorter than that in other thin films made from inorganic nanomaterials, indicating that water evaporation is more efficient in the biofilm. This conclusion is supported by our experimental data showing that the rate of water evaporation from a biofilm was even faster than from an open water surface (Fig. R4). The reduced fluidic length, corresponding to an increased pressure gradient, is consistent with the enhanced energy density in the biofilm devices.

Biofilm-sheet devices achieved a voltage output $\sim 0.4 \text{ V}$ along the vertical thickness, indicating that the fluidic length was even shorter (e.g., $\leq 40 \mu\text{m}$). The possibilities for direct verification are limited because $40 \mu\text{m}$ is the minimal thickness achieved in a single layer (Fig. R5a). This further reduced fluidic length still can be generally understood from the geometric effect. Specifically, the presence of meniscus effect (*J. Mater. Chem. C* 8, 9133-9146, 2020) at the biofilm-substrate interface in the planar device can extend the fluidic length (Fig. R5b).

Fig R3. Open-circuit voltage (V_o) was measured from planer biofilm devices of different electrode spacings.

Fig R4. Measured water evaporation rates from open water (black) and biofilm (gray).

In the revised manuscript (page 3 & 4): “The voltage output declined with a decrease of the electrode spacing (Fig. S7).”... “Water evaporation through the biofilms was even faster than from an open water surface (Fig. S13), which may account for the rapid establishment of the streaming potential over a short distance in the biofilm plane (Fig. S7). A similar trend was maintained across the vertical thickness of biofilms (Fig. S14), consistent with an isotropic transport expected from the 3D distributed porosity in the biofilm (Fig. 1c). The enhanced streaming efficiency and evaporation rate in the biofilm can account for the improved energy density compared to other thin-film devices.^{10,12,18}” **Correspondingly, we have added Fig. R3, R4, R5 and associated discussions to new Supplementary Fig. S7, S13, S14.**

3. Streaming currents are supposed to be carried by migrating ions, but the external circuit uses electrons. So, at some point these ions are transferring their charges to the electrodes. One would expect some change in the biofilm or gas formation depending on the ions involved, especially after month-long current measurements. Is there such evidence?

For the month-long current production (Fig. 1d, main paper), a total charge of ~ 5.3 Coulomb was involved in the flow. Given that the current production is largely independent of the material composition of the biofilms and ionic species/strengths of the solution, water-based redox which can yield gas release (e.g., H_2 , O_2) would be a more reasonable expectation. If the charge transfer leads to gas formation (e.g., H_2), it should yield $\sim 2.7 \times 10^{-5}$ mol of gas, or 0.6 cm^3 volume. The volume corresponds to ~ 1200 bubbles of 1 mm size. So on average, about 2 bubbles should be generated each hour, which could be readily observed. However, we did not observe bubbles during the process.

It is possible that the bubbles were much smaller and dissipated in an unobservable way. However, it is also likely that the generation of external electron current does not require a redox process in the ionic species. One generally adopted understanding of the streaming potential is that the water flow drags the charge flow in the diffuse layer (which is not charge neutral) to build up a charge gradient (Nano Lett. 9, 1984-1988, 2009).

Now consider that these net charges will induce image charges of opposite signs (electron) inside the material, then the external current can be driven by the electron gradient in the material which does not require redox (Fig. R6). This may be a more reasonable explanation, given the generality of streaming current. A recent study adopted a similar image-charge model, although no experimental support was provided (*Angew. Chem. Int. Ed* 59, 10619, 2020). Our observation can provide experimental evidence consistent with the image-charge model.

We still acknowledge that an in-depth understanding and definitive conclusions will require further studies, in conjunction with the fact that the charge mechanism in streaming current is generally unknown to the field even though the phenomenon was discovered over a century ago (*Adv. Mater.* 32, 2003722, 2020).

In the revised manuscript (page 8), we have included following discussion: “Further rational optimization of microbial films for electricity generation will need a better understanding of how the streaming current is generated, which currently is poorly understood.⁹ Based on our experimental evidence, a model that assumes the induction of image electrons in the materials can account for the closed-loop current flow without the involvement of redox process (Fig. S22).”

Correspondingly, we have added all above experimental result, analysis, the schematic model (Fig. R6), and associated mechanistic description in the new **Supplementary Fig. S22 and caption**.

4. Line 92-95: “High salt concentrations diminish the power output of some evaporation-based current generation devices (13) because the decreased Debye length at high ionic strength reduces the double-layer overlap and hence the streaming efficiency (19). However, the biofilm-sheet devices maintained electric outputs in salt solutions with an ionic concentration (0.5 M) higher than that (~0.15 M) in bodily water (Fig. 3a).” This is a good result that demonstrates the capability of their device, but it goes strongly against the claim that the power generation is due to streaming. I recommend authors inspect Figures 2 and 3 of reference 19 that they cite here. If the open circuit voltage and current do not change by orders magnitude at 0.5 M NaCl or KCl (The data in Fig. 3a suggest they do not), then the authors should look for an alternative explanation. These results are hard to reconcile with a streaming current mechanism.

For the additional evidence and analysis described in responses #1 and #2, streaming mechanism is a more reasonable explanation of the electricity generation. It is thus more likely that other factors have contributed to the ionic behavior deviating from that in a standard microfluidic channel.

First, typical microfluidic channels are made from inorganic materials, which have monolithic surface charge groups. This means that even at nanoscale channel size, at least ions of the opposite charge can still easily pass the channel and the local ionic concentration is not substantially reduced. In contrast, biofilms are full of amphiphilic peptide groups (PCCP 17, 22217-22226, 2015). For nanoscale pores (revealed by TEM), both positive and negative ions in the solution may be effectively repelled by the amphiphilic surface to yield effective reduction in the local ionic concentration. This is not uncommon as amphiphilic peptide was employed to coat membranes for improving water desalination (*Appl. Polymer Sci.* 135, 46169, 2018). The analysis is also consistent with previous studies (*Nano Lett.* 15, 2143-2148, 2015) showing that some organic porous material surfaces can increase the effective Debye length (*i.e.*, decrease the local ionic concentration).

Fig R7. Atomic force microscopy image of protein nanowires from *Geobacter* (Mbio 12, e02209, 2021).

Second, the biofilms are made from microorganism *Geobacter* which produce extracellular protein nanowires (3 nm diameter, 2-6 μm long) (Fig. R7). This means that the nanofluidic channels in the biofilm are infiltrated with many protein nanowires. These ultrasmall protein nanowires do not substantially block water transport but introduce a network of water-solid interfaces to effectively reduce the spacing between double layers. As a result, they effectively increase the double-layer overlapping (Fig. R8) to increase the streaming efficiency.

The qualitative analysis provided above, which is based on known properties/studies may provide some reasonable explanation for the mechanism, but we also acknowledge that the details will require further investigation. In the revised manuscript (page 5), we have added following discussion: “This can be attributed to the special material properties in the biofilm. Specifically, unlike many inorganic materials having monolithic surface charge state, the biofilm contains many amphiphilic groups.²¹ The resultant amphiphilic surface may help to effectively repel both positive and negative ions. As a result, the local ionic strength is reduced or the effective Debye length is increased. This effect was observed in other porous organic materials.^{22,23} Meanwhile, the biofilm contains a high density of protein nanowires synthesized by *G. sulfurreducens* for charge transport in the colony.⁶ These ultrasmall nanowires (e.g., 3 nm diameter) do not block water transport but introduce a high-density network of water-solid interfaces to effectively increase the number of double layers (Fig. S17). Together, the increases in both the effective Debye length and number of double layers can help to maintain the double-layer overlap or streaming efficiency at high ionic strength. The results show that the biofilm-sheet devices can be used in diverse aqueous environments for energy harvesting.” Correspondingly, we have added the mechanistic schematic Fig. R8 into a new **Supplementary Fig. S17**.

I believe it would be unfair to the authors to ask them to find out the mechanism of the power generation in one publication. This appears to be a complex problem. The primary result that they are able generate current with an open circuit voltage around 0.4 V is quite interesting and it could stimulate further research towards its origin. However, I would object claims regarding the mechanism without adequate evidence. For example, their claim in the title “generating electricity from water evaporation through microbial biofilms” is not adequately supported in this reviewer’s opinion (as explained above in items 1-4). I agree that water is necessary and electricity is being generated, but the data seems to be going against a streaming current mechanism. Could there possibly be another mechanism that is facilitated by evaporation?

We thank the reviewer for the considerate understanding and accommodation. We hope that our additional efforts have now provided more convincing evidence and consistence in the general conclusion, although (as the reviewer acknowledged) the detailed mechanism warrants further study.

Minor comments:

i. Line 317: the sentence ending with planktonic seems to be incomplete.

We thank reviewer for pointing out the typo. In the revised manuscript, we have corrected the sentence to “Cross-sectional TEM image of a biofilm-mat from filtered *E. coli*. Scale bar, 1 μm .”

ii. Fig. S6: It’s not clear to me what “depletion of water source” means. Do the authors stop water supply and then the level of water gradually goes down? If so, why would the voltage gradually go down? The drop in voltage seems to have an exponentially decaying characteristic. This data could be informative if analyzed quantitatively.

Indeed, we let the water level gradually go down (through evaporation) in a device configuration as shown in Fig. R9. Therefore, the slow decay in voltage was mostly due to the slow drop of the water level (e.g., the pair of electrodes gradually moved out of the effective microfluidic region).

Fig R9. Schematic of testing setup.

Alternatively, when we directly pulled the device out of the water, a much faster decay in the voltage output was observed (Fig. R10). Note that the voltage did not instantly drop because it still required time for the adsorbed water to evaporate.

Fig R10. Voltage decay by pulling the device out the water.

In the revised manuscript, we have now replaced the original Fig. S6 with the new data of Fig. R10 (and description) for better understanding.

iii. Fig. 1a, the schematic on the right side of the panel: What is the grey square around the bottom mesh electrode?

The grey square is a PDMS thin-film substrate that supports the device to float on the water. We have now added additional description in the caption.

iv. Error bars should be defined in all figure panels that present error bars.

Error bars are standard deviations (s.d.) and we have now specified the definition.

Again, we would like to thank the reviewer for the time, effort and valuable suggestions to help us to improve the work. We believe our efforts have improved the presentation and the quality of the publication.

Reviewer #3:

The work demonstrated a creative study showing the benefits of using microbial biomaterials for electricity harvesting from water evaporation (harnessing the ‘streaming current’ mechanism). Several highlights of the work include: 1. Bringing in a concept in harnessing microbial materials for fabricating ‘hydrovoltaic’ devices. The revealed generic effect shows the potential to broadly explore this renewable material source for clean energy, with expected benefits in ease of material production, abundance, environmental friendliness. 2. Achieving exceptional energy density and current sustainability compared to devices/technologies fabricated from traditional engineered materials. 3. The material composition and structure in the microbial film overcome the energy decay (resultant from Debye screening) in salty water observed in conventional materials. It has the implication to push forward the field of microbattery or micro fuel cell (both requiring chemical feed). Given above reasons, I highly recommend this work to be published in *Nat. Commun.* Some suggestions for potential improvements:

We thank the reviewer for confirming the value in the work by nicely summarizing the highlights. Below please find our detailed responses that address each suggestion/question the reviewer raised. To make it clear, we have used *italic* fonts for the reviewers’ comments, **black** fonts for our replies, and **blue** fonts for revisions.

i. Electrode-grown microbial biofilms show better energy efficiency than biofilms made from filtering, due to the collapse of inter-cellular space during filtering. Given that solution/filtering processing may still have the advantage in higher production throughput, is there any way to engineer improved porosity in filtered biofilm to increase energy density?

Electrode-grown biofilms were supported with an extracellular polymeric matrix to yield a high density of 3D nanoporosity (Fig. 1c in main paper), which helped to triple the current generation compared to filtered mats with reduced porosity (Fig. 4c in main paper). Replicating the 3D nanoporosity in electrode-grown biofilm is difficult at this stage. However, we tested the feasibility of incorporating an artificial scaffold in filtered mats to improve the energy generation.

For the proof-of-concept demonstration, the bacteria were infiltrated into a tissue paper featuring 3D porosity to form hybrid mats (Fig. R1a). Both *Geobacter* and *E. coli* were used to show the generality of this approach. The energy outputs (e.g., $\sim V_o \times I_{sc}$) from infiltrated hybrid mats of *Geobacter* and *E. coli* increased $\sim 46\%$ and $\sim 375\%$ compared to values from mats of pure *Geobacter* and *E. coli*, respectively (Fig. R1b).

Fig. R1. (a) Scanning electron microscope images of (left) a tissue paper and (right) a tissue paper infiltrated with *E. coli*. Scale bars, (left) 100 μm, (right) 10 μm. (b) Average V_o (gray) and I_{sc} (red) measured from devices fabricated with biofilm-mats of *Geobacter*, infiltrated *Geobacter*, *E. coli*, and infiltrated *E. coli*.

Although the output was still lower than that from electrode-grown biofilms, the results demonstrate the potential of improving energy efficiency by engineering hybrid/composite biofilms. Other methods, such as culturing bacteria in a scaffold or mixing nanofibers in bacteria during filtering, may offer broad options for composition engineering, but the systematic study of these possibilities is a new direction will require a substantial future effort.

In the revised manuscript (page 7-8), we have added following description: “These results suggested that improving the microstructure of filtered biofilm-mats by adding a supportive scaffold might increase the electricity generation. As proof-of-concept demonstration, planktonic cells were infiltrated within a tissue paper with microporous structure (Fig. 4d). Devices fabricated from these hybrid mats of *G. sulfurreducens* generated an average voltage of 0.35 V and a current of 0.27 μ A (Fig. 4e), or 113% and 129% produced by the same size devices fabricated solely from mats of *G. sulfurreducens*. Devices fabricated from hybrid mats of *E. coli* generated an average voltage of 0.33 V and a current of 0.25 μ A (Fig. 4e), or 165% and 227% produced by the same size devices fabricated solely from mats of *E. coli*. These results indicate that engineering biofilm composites can improve evaporation-based energy production. Future efforts in composite engineering may include directly culturing planktonic cells in a scaffold sheet or filtering solutions of mixed nanofibers and cells.”. Correspondingly, we have added Fig. R1a and Fig. R1b to a new Figs. 4d, e.

ii. The scalability of the device has been well studied in terms of device planar size. What is the performance/scalability with respect to the film vertical thickness? Will the pore size in the electrode still have considerable effect on the results?

Devices fabricated from monolayer (~40 μ m thick), double-layer (~80 μ m thick), and triple-layer (~120 μ m thick) biofilm-sheets were tested. The voltage output slightly increased from 0.4 V to 0.47 V (Fig. R2). It indicates that the effective fluidic length across the biofilm, which is responsible for building up the streaming potential, is very short (e.g., ≤ 40 μ m). This is consistent with our measurement showing that water evaporation rate through the biofilm is even faster than from an open water surface (Fig. R3), indicating that the liquid fluidic phase quickly transitions to vapor phase in the biofilm.

Using lithography, we have reduced the pore size in the mesh electrodes from 1000 μ m to 20 μ m (at an optimal porosity of 0.4). The results showed the current output increased but the voltage output largely maintained (Fig. R4). The results are consistent with the expectation that voltage (streaming potential) is dependent on the fluidic length/thickness (which does not change), whereas a dense grid can improve the charge collection (i.e., reduced sheet resistance).

In the revised manuscript (page 4-5), we have added following description: “Water evaporation through the biofilm was even faster than from the open water surface (Fig. S13), which may account for the rapid establishment of the streaming potential over a short distance in the biofilm plane (Fig. S7). A similar trend was maintained across the vertical thickness of biofilm (Fig. S14), consistent with an isotropic transport expected from the 3D distributed porosity in the biofilm (Fig. 1c). The enhanced streaming efficiency and evaporation rate in the biofilm can account for the improved energy density compared to other thin-film devices.^{10,12,18}” ... “For the quick establishment of streaming potential over a short distance in the biofilm, the voltage output was largely independent of film thickness and a single layer could attain optimal value (Fig. S14).” ... “With a fixed porosity of 0.4, reducing the pore size or increasing the grid density, which is expected to reduce the overall sheet resistance in the device, further improved the current collection (Fig. 2d).” Correspondingly, we have added Fig. R2, R3, R4 into new Fig. S14, S13, Fig. 2d.

Fig R4. V_o (black) and I_{sc} (red) from devices with fixed porosity of 0.4 but varying pore sizes (from 1000 to 20 μm). The device size was 25 mm².

iii. It would be useful to see if the devices can work with sea water for broader potential.

We followed the reviewer’s suggestion and tested the electric outputs from the biofilm-sheet device in artificial seawater (475.7 mM NaCl, 10.8 mM CaCl₂, 25.6 mM MgCl₂·6H₂O, 28.2 mM MgSO₄). The results showed that the device maintained electric outputs (e.g., >50% of values in water) in salt solutions and seawater, compared to diminished outputs (e.g., <5% of value in water) in previously described evaporation-based devices.

Fig. R5. (a, b) Open-circuit voltage (V_o) and short circuit current (I_{sc}) measured a biofilm-sheet device pleased on artificial seawater. (c) Statistical comparison of device performance in different aqueous environments (DI water, 0.5 M NaCl, 0.5 KCl, and artificial seawater).

In the revised manuscript (page 5-6), we have included the above results and discussion of the mechanistic cause: “High salt concentrations diminish the electric output (e.g., <5% of value in water) of typical evaporation-based current generation devices^{10,12,18} because the decreased Debye length at high ionic

strength reduces the double-layer overlap and hence the streaming efficiency.²⁰ However, the biofilm-sheet devices maintained electric outputs (*e.g.*, >50% of values in water) in salt solutions and seawater with an ionic concentration of 0.5 M. (Fig. 3a). This can be attributed to special material properties of the biofilm. Specifically, unlike many inorganic materials, which have monolithic surface charge state, the biofilm contains many amphiphilic groups.²¹ The resultant amphiphilic surface may help to effectively repel both positive and negative ions. As a result, the local ionic strength is reduced or the effective Debye length is increased. This effect was observed in other porous organic materials.^{22,23} Furthermore, the biofilm has a high density of protein nanowires synthesized by *G. sulfurreducens* for charge transport.⁶ These ultras-small nanowires (*e.g.*, 3 nm diameter) do not block water transport but introduce a high-density network of water-solid interfaces to effectively increase the number of double layers (Fig. S17). Together, the increases in both the effective Debye length and number of double layers can help to maintain the double-layer overlap or streaming efficiency at high ionic strength. These results show that the biofilm-sheet devices can be used in diverse aqueous environments for energy harvesting.” **We have correspondingly added Fig. R5 into Fig. 3a.**

Again, we would like to thank the reviewer for the time, effort and valuable suggestions to help us to improve the work. We believe our efforts have improved the presentation to the quality of publication.

REVIEWERS' COMMENTS

Reviewer #1 (Remarks to the Author):

In the revised manuscript and rebuttal letter, the authors thoroughly discussed and addressed the questions raised in the first-round of review. Publication in Nature Communication is highly recommended!

Reviewer #2 (Remarks to the Author):

The authors have thoroughly revised their manuscript and added new results that meaningfully added to the issues about understanding of the power generation mechanism. I appreciate the discussions that relied on theoretical models of streaming potentials and currents, and the new experiments that tested some predictions that derive from the theory. I recommend publication of the manuscript. However, I want to comment on the "image charge" mechanism the authors proposed in the revision. Because they frame their model as a proposal and resolving this issue is not necessary for this publication, I am OK if they do not make any changes in response to my comment below.

Comment: The authors' proposed mechanism relies on the gradients in electron density in the channel material, as induced by the charge gradients of ions dragged by the flow. The gradients in electrons, according to the proposed mechanism, would then sustain a drift current in an external circuit. I would like to direct the authors' attention to the balance in power input and output to this system. Because there is a current in the external circuit, there is steady power output (they demonstrate applications where this power is used). Now, because of energy conservation, an equal amount of power must be input to the circuit. Based on the authors' proposal, the power has to come from the fluid flow by mechanical means. Mechanical power is given by the product of force and velocity (alternatively, by the product of pressure drop and volumetric flow rate). If the charged particles have a steady distribution (i.e., there is no net flow of ions), their net velocity must be zero (the drift cancels the diffusion). So, I cannot see how power is transferred to electrical circuit in the proposed mechanism.

Reviewer #3 (Remarks to the Author):

I think the authors have taken the reviewers' comments well to strengthen the paper, and have done a fairly thorough revision on the manuscript. Again, this is a very nice piece of work, and should be accepted soon for publication.

Response to Reviewers

To make it clear, we have used *italic* fonts for the reviewers' comments, **black** fonts for our replies, and **blue** fonts for revisions.

Reviewer #1:

Reviewer #1 (Remarks to the Author):

In the revised manuscript and rebuttal letter, the authors thoroughly discussed and addressed the questions raised in the first-round of review. Publication in Nature Communication is highly recommended!

We greatly thank the reviewer for the effort and help in improving our research work!

Reviewer #2:

The authors have thoroughly revised their manuscript and added new results that meaningfully added to the issues about understanding of the power generation mechanism. I appreciate the discussions that relied on theoretical models of streaming potentials and currents, and the new experiments that tested some predictions that derive from the theory. I recommend publication of the manuscript. However, I want to comment on the "image charge" mechanism the authors proposed in the revision. Because they frame their model as a proposal and resolving this issue is not necessary for this publication, I am OK if they do not make any changes in response to my comment below.

We greatly thank the reviewer for the effort and help in improving our research work. We are opt to keep the qualitative model for reasons as briefly described below.

Comment: The authors' proposed mechanism relies on the gradients in electron density in the channel material, as induced by the charge gradients of ions dragged by the flow. The gradients in electrons, according to the proposed mechanism, would then sustain a drift current in an external circuit. I would like to direct the authors' attention to the balance in power input and output to this system. Because there is a current in the external circuit, there is steady power output (they demonstrate applications where this power is used). Now, because of energy conservation, an equal amount of power must be input to the circuit. Based on the authors' proposal, the power has to come from the fluid flow by mechanical means. Mechanical power is given by the product of force and velocity (alternatively, by the product of pressure drop and volumetric flow rate). If the charged particles have a steady distribution (i.e., there is no net flow of ions), their net velocity must be zero (the drift cancels the diffusion). So, I cannot see how power is transferred to electrical circuit in the proposed mechanism.

We thank the reviewer for bringing about the attention. We respectfully think that the reviewer may have missed one link. Specifically, maintaining the charge gradient (not the image charge gradient) in the channel requires the continuous water flow. Thus, mechanical work needs to be done to this charge layer (i.e., the charge layer induces additional drag to the flow). This consists the source of initial energy input (from the mechanic work of flow).

Reviewer #3

I think the authors have taken the reviewers' comments well to strengthen the paper, and have done a fairly thorough revision on the manuscript. Again, this is a very nice piece of work, and should be accepted soon for publication.

We greatly thank the reviewer for the effort and help in improving our research work!